# Anti-Quorum Sensing Activities of Gliptins against *Pseudomonas aeruginosa* and *Staphylococcus aureus*

**DOI:** 10.3390/biomedicines10051169

**Published:** 2022-05-18

**Authors:** Maan T. Khayat, Hisham A. Abbas, Tarek S. Ibrahim, Ahdab N. Khayyat, Majed Alharbi, Khaled M. Darwish, Sameh S. Elhady, El-Sayed Khafagy, Martin K. Safo, Wael A. H. Hegazy

**Affiliations:** 1Department of Pharmaceutical Chemistry, Faculty of Pharmacy, King Abdulaziz University, Jeddah 21589, Saudi Arabia; tmabrahem@kau.edu.sa (T.S.I.); ankhayyat@kau.edu.sa (A.N.K.); maaalharbi1@kau.edu.sa (M.A.); 2Department of Microbiology and Immunology, Faculty of Pharmacy, Zagazig University, Zagazig 44519, Egypt; hishamabbas2008@gmail.com; 3Department of Medicinal Chemistry, Faculty of Pharmacy, Suez Canal University, Ismailia 41522, Egypt; khaled_darwish@pharm.suez.edu.eg; 4Department of Natural Products, Faculty of Pharmacy, King Abdulaziz University, Jeddah 21589, Saudi Arabia; ssahmed@kau.edu.sa; 5Department of Pharmaceutics, College of Pharmacy, Prince Sattam Bin Abdulaziz University, Al-Kharj 11942, Saudi Arabia; e.khafagy@psau.edu.sa; 6Department of Pharmaceutics and Industrial Pharmacy, Faculty of Pharmacy, Suez Canal University, Ismailia 41552, Egypt; 7Department of Medicinal Chemistry, School of Pharmacy, Virginia Commonwealth University, Richmond, VA 23219, USA; msafo@vcu.edu; 8Department of Pharmaceutical Sciences, Pharmacy Program, Oman College of Health Sciences, Muscat 113, Oman

**Keywords:** antivirulence, bacterial resistance, gliptins, quorum sensing, industrial development

## Abstract

The development of bacterial resistance to traditional antibiotics constitutes an emerging public health issue. Promising approaches have been innovated to conquer bacterial resistance, and targeting bacterial virulence is one of these approaches. Bacterial virulence mitigation offers several merits, as antivirulence agents do not affect the growth of bacteria and hence do not induce bacteria to develop resistance. In this direction, numerous drugs have been repurposed as antivirulence agents prior to their clinical use alone or in combination with traditional antibiotics. Quorum sensing (QS) plays a key role in controlling bacterial virulence. In the current study, dipeptidase inhibitor-4 (DPI-4) antidiabetic gliptins were screened for their antivirulence and anti-quorum sensing (anti-QS) activities against Gram-negative *Pseudomonas aeruginosa* and Gram-positive *Staphylococcus aureus*. Upon assessing their antibiofilm activities, the ten tested gliptins significantly diminished biofilm formation. In particular, sitagliptin exhibited the most efficient antibiofilm activity, so it was chosen as a representative of all gliptins to further investigate its antivirulence activity. Sitagliptin significantly protected mice from *P. aeruginosa* and *S. aureus* pathogenesis. Furthermore, sitagliptin downregulated QS-encoding genes in *P. aeruginosa* and *S. aureus*. To test the anti-QS activities of gliptins, a detailed molecular docking study was conducted to evaluate the gliptins’ binding affinities to *P. aeruginosa* and *S. aureus* QS receptors, which helped explain the anti-QS activities of gliptins, particularly sitagliptin and omarigliptin. In conclusion, this study evaluates the possible antivirulence and anti-QS activities of gliptins that could be promising novel candidates for the treatment of aggressive Gram-negative or -positive bacterial infections either alone or as adjuvants to other antibiotics.

## 1. Introduction

Since their discovery, antibiotics have revolutionized the treatment of infectious diseases; however, this great achievement has been fading due to the cumulative ubiquity of microbial resistance, constituting a major global health challenge [1,2]. With the dwindling innovation of new effective chemotherapeutics, death due to untreatable infectious diseases are increasing and may reach ten million by 2050 [1,3]. Bacteria have developed resistance to almost all antibiotic classes [4,5,6,7]. This situation mandates the invention of new approaches to overcome bacterial resistance [8,9,10,11]. Quenching bacterial virulence has been proven to be among the most effective approaches to decreasing the development of bacterial resistance [4,12,13,14,15]. This approach leads to the waning of bacterial pathogenesis, allowing for the easy eradication of the bacterial infection by the host immune system without affecting bacterial growth, preventing the development of bacterial resistance [12,16,17,18,19]. Furthermore, targeting the virulence factors in pathogenic bacteria does not destroy the normal bacterial flora [12,20].

Bacteria recruit an extensive panel of virulence factors and employ diverse systems to regulate the expression of these factors. The bacterial arsenal of virulence factors extends from acquiring bacterial structures, such as capsules, pili, flagella, and others, to the production of a diverse array of destructive extracellular enzymes, such as protease, elastase, urease, hemolysins, and others [5,8,21]. Bacterial systems such as secretion and quorum sensing (QS) systems orchestrate the production of these virulence factors in a sophisticated manner that ensures the establishment of bacterial infection in the host tissues and cells [22,23]. Cumulative evidence has drawn attention to the crucial QS roles in controlling numerous bacterial virulence factors, such as biofilm formation, motilities, and the production of numerous extracellular enzymes [12,22,23,24]. QS is the chemical language that bacteria use in their communication in an inducer/receptor manner to regulate their invasion and accommodation in the host cell niche [12]. Both Gram-negative and -positive bacteria utilize QS systems to regulate their pathogenesis. For instance, Gram-negative bacteria often employ several autoinducers (AIs), mostly acyl-homoserine lactones (AHLs), which diffuse through bacterial cell membranes to be recognized by QS receptors, mainly LuxR types, on the surface of other bacterial cells [12,22]. Following that, the LuxR–AHL complexes bind to short sequences of DNA located upstream of virulence genes, known as lux boxes, regulating the expression of these genes [25,26,27]. In Gram-positive bacteria, the QS systems use cytoplasmic transcription factors and sensor kinase receptors to sense oligopeptides in order to regulate the expression of virulence genes [12,26]. Meanwhile, bacteria evoke all of these virulence factors under the control of QS systems, and targeting bacterial QS systems could diminish their virulence without any influence on bacterial growth so as to not induce bacterial resistance development [17,25,28].

Several tactics have been suggested to target bacterial virulence by utilizing a wide array of natural and synthesized chemical moieties [10,11,29,30,31,32,33,34], including the use of approved or repurposed drugs. Approved drugs are increasingly being repurposed for new therapeutic applications [25,35]. The assured safety of repurposed drugs shortens the time and minimizes the cost for their development [35,36]. Nonetheless, further pharmaceutical and pharmacological investigations are required to test the effectiveness of new clinical applications [13,37,38]. With this in mind, diverse drugs have been screened for their antivirulence activities prior to their repurposing [13,14,16,18].

Gliptins, a novel class of antidiabetics, are dipeptidyl peptidase-4 inhibitors (DPI-4) for improving β-cell health and controlling blood glucose levels in diabetes mellitus type 2 [13]. The gliptin class comprises several members of nitrogenous heterocyclic compounds. Sitagliptin was the first gliptin approved as an oral antidiabetic [13,25,39,40]. In previous studies, sitagliptin mitigated *Serratia marcescens* virulence factors, such as biofilm formation, bacterial motility, and the production of protease, catalase, and hemolysins, and diminished its pathogenesis in vivo [17,39]. This led us to screen the antivirulence activity of other gliptins.

In this study, we preliminarily investigated the ability of different gliptins to modulate both Gram-positive and -negative QS systems to determine whether they could potentially serve as antivirulence agents and be employed as adjuvants to traditional antibiotics to overcome bacterial resistance. The antibiofilm activities of ten gliptins were evaluated against Gram-positive *Staphylococcus aureus* and Gram-negative *Pseudomonas aeruginosa*. Furthermore, the antivirulence activity of the most promising candidate, sitagliptin, was evaluated in vivo. The effect of sitagliptin on the expression of QS-encoding genes in *S. aureus* and *P. aeruginosa* was assessed to evaluate its anti-QS activity. A detailed in silico molecular docking study was conducted to understand the mechanism underlying the anti-QS activities of gliptins. The present findings shed light on the possible application of gliptins as promising antivirulence agents; however, further detailed pharmacological and toxicological studies are required prior to clinical application to avoid any undesired side effects. 

## 2. Materials and Methods

### 2.1. Bacterial Strains, Microbiological Media, and Chemicals

Gliptins’ antivirulence and antibiofilm activities were evaluated using model strains *P. aeruginosa* (ATCC 27853) and *S. aureus* (ATCC 6538). Microbiological media, Mueller Hinton (MH) and Luria–Bertani (LB) broth or agar, and tryptone soy broth (TSB) were obtained from Oxoid (Hampshire, UK). All gliptins were purchased from Cayman Chemicals (Ann Arbor, MI, USA). Dimethyl sulfoxide (DMSO) was used to dissolve the gliptins, and the DMSO and other chemicals were of analytical grade.

### 2.2. Detection of Minimum Inhibitory Concentrations (MICs) of Gliptins

The broth microdilution method was used to detect the MICs of gliptins against tested strains according to the Clinical Laboratory and Standards Institute Guidelines (CLSI, 2020) [14,19].

### 2.3. Assessment of Gliptins’ Effects on Bacterial Growth

The antivirulence and antibiofilm activities of gliptins at their MIC concentrations could affect bacterial growth and, in turn, affect bacterial virulence. To exclude any effect of gliptins on bacterial growth, the antivirulence activities of tested gliptins were tested at sub-MIC (1/5 MIC) concentrations. For more confidence, the effects of gliptins at sub-MIC (1/5 MIC) concentrations were assessed as described previously [7,25,39]. Importantly, all of the gliptins’ antivirulence and antibiofilm activities were evaluated at their sub-MICs (1/5 MIC).

### 2.4. Evaluation of Antibiofilm Activities of Gliptins

Well-known documented strong biofilm-forming bacterial strains *Pseudomonas aeruginosa* (ATCC 27853) and *Staphylococcus aureus* (ATCC 6538) were employed to assess the antibiofilm abilities of different gliptins [3,41,42]. The crystal violet method was used to evaluate the gliptins’ inhibitory activities on biofilm formation, as indicated previously [10,11,25]. The antibiofilm effects of tested gliptins are presented as a percentage change from untreated bacterial controls.

### 2.5. Visualization of the Biofilm Formation Inhibition

For electron microscopic scanning analysis of the formed biofilms in the presence and absence of gliptins at sub-MICs, JOEL JSM-6510LV Scanning Electron Microscope was used (Boston, MA, USA) [43]. The biofilms of the tested strains were formed on glass coverslips placed in polystyrene Petri plates in the presence and absence of sub-MICs of tested gliptins. The plates were incubated for 24 h at 37 °C, and the coverslips were washed with water three times and stained with crystal violet (1%) for 20 min [3,39]. For electron microscopic scanning analysis, biofilms formed on glass coverslips were fixed by glutaraldehyde (2.5%) for 120 min. After washing with distilled water, the coverslips were dehydrated by increasing concentrations of ethanol (50%, 60%, 70%, 80%, 90%, and 100%) for 30 s. The samples were gold coated after critical-point drying and examined under scanning electron microscope.

### 2.6. Quantification of QS- and Virulence-Encoding Gene Expression

To explore the influence of gliptins on the expression of QS- and virulence-encoding genes in *P. aeruginosa* and *S. aureus* strains, RNA was extracted from tested strains’ cultures treated or untreated with tested gliptins using RNeasy Mini kit (Qiagen, Hilden, Germany). Then, NanoDropTM 2000 (Thermo Fisher Scientific, Waltham, MA, USA) was employed to quantify the extracted RNA [18,21,44]. The extracted RNA was used to synthesize cDNA using a Reverse Transcriptase kit (Thermo Fisher Scientific, Waltham, MA, USA), and SYBR Green Quantitative RT-qPCR Kit (Thermo Fisher Scientific, Waltham, MA, USA) was used to amplify the obtained cDNA. The expression levels of the tested genes were normalized relative to the critical threshold (CT) mean values of housekeeping gene *rpoD* for *P. aeruginosa* virulence-encoding genes and 16s *rRNA* for *S. aureus* virulence-encoding genes using the 2^−∆∆Ct^ method [45,46]. The used primers in this study are listed in Table 1.

### 2.7. In Vivo Protection Assay

A mouse survival test was used to evaluate the ability of the tested gliptins to protect mice against *P. aeruginosa* and *S. aureus* strains, as described previously [18,19,45,53]. Briefly, 1 × 10^8^ CFU/mL of tested strains treated or not with gliptins at sub-MICs were prepared in phosphate-buffered saline to be injected into the mice. Ten mice groups, each comprising five healthy female albino mice with the same wight and at the same age (3-weeks old), were established to evaluate the protective ability of the tested gliptins. Five groups were assigned for evaluation of in vivo protection against *P. aeruginosa*; three negative control groups were not injected with bacteria or were intraperitoneally (ip) injected with 100 μL of sterile PBS or DMSO. Meanwhile, one positive group was ip injected with 100 μL of untreated *P. aeruginosa*, and the fifth test group was injected with 100 μL of *P. aeruginosa* treated with gliptin at sub-MIC. The same was performed for assessment of the antivirulence activity against *S. aureus*: three negative control groups and one positive control group in addition to one test group, which was injected with *S. aureus* treated with gliptin at sub-MIC. All mice were housed in suitable cages with normal conditions of aeration and feeding, and the mice’s survival in all groups was documented daily over five days.

### 2.8. Ligand Construction, Target Preparation, and Docking Protocol

The designated gliptins and reference ligands of the *P. aeruginosa* and *S. aureus* target proteins were constructed via the builder module within the Molecular Operating Environment^®^ MOE2019.01 software package (Chemical Computing Group^TM^, Montreal, QC, Canada). The ligands’ respective isomeric/canonical SMILES strings, obtained from the PubChem database, were utilized to build the ligands in a 3D representation fashion. Subsequently, each constructed ligand was energy-minimized at Merck Molecular ForceField (MMFF94s) partial charges and MMFF94s-modified forcefield through a conjugate-gradient approach of 2000 steps until reaching a root-mean-square (RMS) gradient convergence of 1 × 10^−3^ Kcal/mol/Å2 [54,55,56]. 

Biological targets, on the other hand, were obtained from the deposited 3D crystallographic files at the RCSB-Protein Data Bank for the atomic structure of *P. aeruginosa* LasI-type (PDB ID: 1RO5), QscR (PDB ID: 6CC0), and LasR (PDB ID: 6MVN), as well as the *S. aureus* TCS system: AgrC (PDB ID: 4BXI) and AgrA (PDB ID: 3BS1). Downloaded PDB files were structurally prepared through MOE2019.01 3D protonation (pH 7.4, 300 K temperature, and 0.1 M salt solution in Generalized Born/Volume-Integral implicit solvent model), as well as autocorrection of atom types, partial charges, and bond connectivity. The ATP lid of *S. aureus* AgrC histidine kinase ATP-binding site was missing from the crystalline PDB file, so it was modeled via the User Template module at the SWISS-MODEL server (https://swissmodel.expasy.org/; accessed on 17 October 2021) using the Uniprot entry (A0A1E8WWT6; https://www.uniprot.org/; accessed on 17 October 2021) and the crystalline PDB file (4BXI) as the target sequence file and template file, respectively.

The binding site for each target was defined by the MOE Alpha Site Finder geometrical approach while being refined to include the crucial residues reported in current literature, as thoroughly described within the manuscript. Docking workflow was performed through the induced-fit docking protocol, allowing significant flexibility of pocket residues to identify significant hits exhibiting more negative docking energies (Kcal/mol). Ligand conformations were developed On-the-Fly via bond rotation/ligand placement technique within the defined active site and guided via triangular-matcher protocol [57]. The obtained conformations were ranked via the London_dG scoring system, and the top ten docked poses/ligands were refined and energy-minimized within the target pockets, where only the protein’s sidechain residues were tethered within the forcefield configuration options. Refined poses were then rescored using Generalized Born-Solvation-VI/Weighed Surface-Area_dG forcefield, relying on explicit solvation electrostatics, current-loaded charges, exposure-weighted surface area, and Coulombic electrostatics via protein–ligand van der Waals scores [58,59]. High docking scores, RMSD values below 2.0 Å cut-off, and/or significant interactions with reported crucial pocket residues were considered to select the best ligand’s docking pose. Visual inspection and protein–ligand interaction analysis for the furnished docking poses were achieved by using PyMol2.0.6 Graphical Visualization Software (Schrödinger^TM^, New York, NY, USA) and MOE tools [60]. Redocking of the co-crystallized ligand was carried out through the same docking protocol for validation purposes.

### 2.9. Statistical Analysis

All experiments were performed in triplicate, and the data are presented as mean ± SD. Two-way or one-way ANOVA tests were employed to evaluate the statistical significance, where *p* < 0.05 was considered significant. All statistical outputs, both analysis and graphical, were generated by GraphPad Prism software (Version 8, GraphPad Software Inc., San Diego, CA, USA).

## 3. Results and Discussion

### 3.1. Determination of Minimum Inhibitory Concentrations (MICs) of Gliptins

The MICs of the tested gliptins against *S. aureus* and *P. aeruginosa* were determined using the microtiter broth dilution method [14,60]. The MICs of the tested gliptins are summarized in Table 2.

To evaluate the antibiofilm and antivirulence activities of the tested gliptins, their effects on *P. aeruginosa* or *S. aureus* were assayed at sub-MIC concentrations. The evaluation of the antivirulence activities of tested gliptins at sub-MICs excludes any effect on bacterial growth, and hence, the diminishing of bacterial virulence is not due to inhibiting bacterial growth but only to the gliptins’ direct effects on bacterial virulence machinery [10,16,39]. To ensure the absence of any effects of gliptins (at sub-MICs) on *P. aeruginosa* or *S. aureus* growth, bacterial growth was evaluated in the presence or absence of tested gliptins at sub-MICs (1/5 MIC) (Figure 1). Our findings showed that there was no significant difference between the optical density of *P. aeruginosa* or *S. aureus* in the presence and absence of tested gliptins. Consistently, when bacterial cell counts were performed, there was no significant difference between bacterial counts in the presence and absence of tested gliptins (Appendix A). Subsequent testing of the gliptins was performed at sub-MICs (1/5 MIC) to confirm their antibiofilm or antivirulence activities.

### 3.2. Antibiofilm Activities of Gliptins

Bacterial biofilms are architectural colonies in which bacterial cells adhere to one another and to a static living or non-living surface. The bacterial biofilm matrix is basically composed of polysaccharides, secreted proteins, and extracellular DNAs [2,37,61,62]. Bacterial biofilms are pathogenic and usually associated with chronic and nosocomial infections [37,53]. Furthermore, biofilm formation greatly enhances bacterial resistance and complicates the eradication of microbial infections [12,39]. Thus, the development of antibiofilm agents to be used systemically or even topically will vastly augment the effects of traditional antimicrobials [7,24,36]. In the current work, the antibiofilm activities of gliptins were evaluated by employing the crystal violet method [39,45,53]. Interestingly, all of the tested gliptins (at sub-MICs) diminished the formation of biofilms by *P. aeruginosa* or *S. aureus* (Figure 2). Gliptins diminished *S. aureus* biofilm formation by 28% to 42% and *P. aeruginosa* biofilms by 30 to 88%. In particular, sitagliptin, omarigliptin, and linagliptin showed the most efficient antibiofilm activities against both *P. aeruginosa* and *S. aureus*. Electron microscope images were used to capture the efficient inhibition of bacterial biofilm formation by the gliptins. Representative electron microscope images of *P. aeruginosa* or *S. aureus* exposed to sitagliptin at sub-MIC are shown in Figure 2. Obviously, sitagliptin decreased the numbers of adherent bacterial cells and diminished the biofilms formed by *P. aeruginosa* or *S. aureus*. These findings obviously indicate the possible abilities of these gliptins and related chemical moieties to serve as efficient antibiofilm agents.

### 3.3. In Vivo Antivirulence Activities of Gliptins

In order to evaluate the antivirulence activities of gliptins, an in vivo protection assay was used [16,25,39,53]. Sitagliptin is a leading gliptin member and is widely used alone or in combination with other drugs in diabetes treatments. Sitagliptin showed the highest inhibition of *P. aeruginosa* or *S. aureus* biofilm formation in comparison to other tested gliptins. Furthermore, sitagliptin showed a significant ability to mitigate *Serratia* spp. virulence [29,39]. Based on that, sitagliptin was chosen to be further investigated for its antivirulence as a representative of other gliptins. *P. aeruginosa* is Gram-negative bacteria and is equipped with a considerable arsenal of virulence factors that enable it to invade almost all body tissues [16,20,63]. *S. aureus* Gram-positive cocci is also associated with a wide range of serious skin, eye, wound, and nosocomial infections [7,37]. Both *P. aeruginosa* and *S. aureus* are abundant known human pathogens that are listed in the ESKAPE serious pathogens list (*Enterococcus faecium*, *S. aureus*, *Klebsiella pneumoniae*, *Acinetobacter baumannii*, *P. aeruginosa*, and *Enterobacter* spp.) [10,16]. Thus, the two clinically important pathogens *P. aeruginosa* and *S. aureus*, with quite different virulence behaviors, are good bacterial models to test sitagliptin’s antivirulence activity.

The in vivo protection study of sitagliptin at sub-MIC against *P. aeruginosa* and *S. aureus* was conducted with five mice groups for each bacterial strain (Figure 3). Two negative control groups were established, in which mice were uninjected or intraperitoneally injected with sterile PBS. As two positive controls, mice were injected with untreated bacterial strains or DMSO-treated bacteria. The fifth group was injected with the bacterial strain treated with sitagliptin at sub-MIC. Death was recorded for five days and plotted using the Kaplan–Meier method [10,11,16]. Two mice out of five survived in the groups that were injected with untreated *P. aeruginosa* or DMSO-treated *P. aeruginosa*. Meanwhile, sitagliptin at sub-MIC protected all mice against *P. aeruginosa*. Two deaths were observed among the mice groups that were injected with untreated *S. aureus* or DMSO-treated *S. aureus*. Sitagliptin at sub-MIC protected all mice in the group injected with *S. aureus* treated with sitagliptin. Thus, sitagliptin at sub-MIC conferred 60% and 40% protection against *P. aeruginosa* and *S. aureus*, respectively. It is worth noting that there were no deaths observed in the negative control mice groups. In summary, sitagliptin at sub-MIC significantly lessened the capacity of *P. aeruginosa* and *S. aureus* to kill mice (*p* = 0.0028 and 0.0244, respectively), with the log-rank test used to detect any trends.

### 3.4. Gliptins’ Antivirulence Activities Could Be Due to Interference with QS

#### 3.4.1. Gliptins Downregulate QS-Encoding Genes

Bacterial communities utilize a special chemical language in an autoinducer–receptor manner to orchestrate their virulence and accommodation in the host tissue; this language is QS [12,23,39]. QS regulates biofilm formation, bacterial motility, and the production of virulent exocellular enzymes and pigments [12,23]. This makes QS a good target to mitigate bacterial virulence, leading to investigations involving several approaches and diverse chemical moieties [13,22,23,62]. Both Gram-negative and -positive bacteria utilize QS to regulate their virulence, where they synthesize autoinducers (AIs) to bind to their cognate receptors on the bacterial cell surface [12,23]. It is worth noting that different bacterial species utilize different AIs and receptors [12,64]. For instance, *P. aeruginosa* basically employs three QS systems to regulate its virulence. Two of the systems are LuxI/LuxR types, namely, LasI/LasR and RhlI/RhlR, which sense C_12_-homoserine lactone and butanoyl homoserine lactone autoinducers, respectively [14,16,25]. The third is the non-LuxI/LuxR Pseudomonas quinolone signal (PQS) system, which is encoded by *pqsA*, *B*, *C*, *D*, and *H* [10,14,25]. *P. aeruginosa* utilizes an additional orphan LuxR homolog QscR that binds to LasI autoinducers [12,25]. In the current study, the effect of sitagliptin (as a representative of gliptins) on the expression of the three *P. aeruginosa* QS-encoding genes was evaluated. Sitagliptin significantly downregulated the expression of the autoinducer synthetase-encoding genes *lasI*, *rhlI*, and *pqsA* and also downregulated the genes that encode their cognate receptors *lasR*, *rhlR*, and *pqsR*, respectively (Figure 4A).

*S. aureus*, on the other hand, utilizes a canonical two-component QS system that is encoded by the *agrA-D* locus, which comprises four components and is controlled by the P2 promoter [7,12]. The *agrD* gene encodes the pro-AI, which is processed to the active AI and secreted by the AgrB transmembrane transporter. The secreted AI binds to membrane-bound histidine kinase AgrC, which autophosphorylates and transfers its phosphate group to the response regulator AgrA. Finally, phosphorylated AgrA binds to the P2 promoter, which auto-induces the expression of the *agr* operon [12]. *S. aureus* utilizes oligopeptide AIs that are diverse in their structures and sequences [12,51]. However, for the QS that regulates biofilm formation and virulence in *S. aureus*, the *sarA* global regulator is essential for biofilm formation and regulates several virulence factors [51]. It has been reported that the sigB system is associated with the *S. aureus* regulatory network and increases *sarA* gene expression to enhance its virulence [49,50]. Furthermore, *S. aureus* expresses the *ica* operon to form biofilms and slime layers. In particular, *icaA* is frequently detected in *S. aureus* isolated from catheter samples obtained from hospitalized patients [50,52]. Besides the *ica* operon, fibronectin-binding protein (FnbA) plays a crucial role in the accumulation step in biofilm formation [49,50]. In this study, sitagliptin significantly downregulated the *S. aureus* QS-encoding gene *agrA*, global virulence regulator genes *sarA* and *sigB*, and genes involved in biofilm formation *icaA* and *fnbA* (Figure 4B).

#### 3.4.2. Multi-Target Docking Analysis on QS Receptors

##### Docking Simulation on *P. aeruginosa* Virulence-Regulating Biotargets

To gain more insight into the antipseudomonal activity of the promising gliptins (Figure 5), a validated molecular docking simulation was performed. The in silico study permitted the reliable investigation of the potential affinity of the active gliptins towards three *P. aeruginosa* biological targets regulating the microorganism virulence genes as compared to reported inhibitors. The *P. aeruginosa* LasI-type (PDB ID: 1RO5) acyl-homoserine-lactone (AHL) synthase protein and QscR (PDB ID: 6CC0) and LasR (PDB ID: 6MVN) quorum-sensing (QS) transcription factors were used for this molecular docking study. The 23.11 kDa LasI-type AHL synthase structure has been solved in its monomeric form at 2.30 Å atomic resolution without a bound ligand [65]. The overall architecture of the LasI-type AHL synthase comprises a three-layered α-helix/β-sheet/α-helix sandwich platform with a prominent V-shaped active site presenting an elongated cleft/tunnel for accommodating the substrates used in N-3′oxo-octanoyl-L-homoserine lactone (3-O-C_12_-AHL) synthesis (Figure 6A). The elongated tunnel is lined with several conserved residues across different AHL synthases that permit the proper orientation of the binding substrates for catalysis [66]. LasI-type AHL synthase preferentially produces longer-acyl-chain AHL than any other synthases, consistent with its much-elongated cleft, allowing longer acyl chains to have intrinsically higher binding affinities [65].

The quorum-sensing transcription proteins QscR (55.19 kDa) and LasR (53.90 kDa) crystallize as homodimers and are solved at 2.50 Å and 2.20 Å, respectively (Figure 6B,C). The QscR and LasR structures are in complexes with their respective C_12_-homoserine lactone (C_12_-HSL) and 3-O-C_10_-HSL; these compounds exhibit agonistic activities at EC50 values of 5 nM and 15 nM [67,68]. The overall QscR and LasR structures are quite similar to each other. Both comprise N-terminal α-helix/β-sheet/α-helix-sandwiched ligand-binding domains, yet only QscP was co-crystallized with the *C*-terminal α/β-loop/α DNA-binding domain. Ligands appear completely embedded in their respective proteins with little solvent exposure. The lactone heads of the ligands are anchored at a small inner subsite and stabilized by hydrogen bond interactions as well as hydrophobic contacts. The ligand’s amide linker settles in proximity to several polar residues, permitting hydrogen bond interactions with the pocket’s residues. Finally, the terminal acyl tails of the ligands are directed deep into a large-sized hydrophobic subsite on the other side of the protein’s binding pocket.

##### Docking Analysis at LasI-Type AHL Synthase Binding Site

The performed molecular docking simulation with the in vitro active antipseudomonal gliptins, sitagliptin (SIT) and omarigliptin (OMR), using *P. aeruginosa* LasI synthase showed interesting findings. The ligands illustrated favored orientations at the V-shaped and elongated tunnel of the LasI binding site. Both SIT and OMR showed their respective fluorinated phenyl group nestled at the V-shaped cleft of the binding site while anchoring their fused heterocyclic scaffold in the elongated tunnel pocket (Figure 7A,B). Anchoring of the fluorinated aromatic rings at the V-shaped cleft, rather than the more polar triazole- and pyrazole-based rings, was correlated with the more energetic binding states of either SIT or OMR. The narrow pocket-turn at the V-shaped cleft site imposed greater steric hindrances for the less flexible fused triazole- and pyrazole-based rings as compared to the fluorinated phenyl groups linked with a significantly flexible rotatable bond. Additionally, insertion of the polar triazole- and pyrazole-based rings into the elongated pocket was relevant to satisfy the hydrophilic properties of a few lining residues, including the NH ε sidechain of Trp69 and the electrophilic oxygen atom at the Thr121 sidechain. Placing the ligand’s fluorinated aromatic rings at the V-shaped cleft permitted close proximity to the lining residues, Arg30, Trp33, and Phe27, favoring respective polar (hydrogen binding) and π-mediated hydrophobic contacts with the residue sidechains (Appendix A).

The significance of polar contacts with Arg30 was highlighted by Gould et al., who explained the substrate specificity of LasI as compared to other AHL synthases, e.g., EsaI. The authors modeled 3-O-C_12_-acylphosphopantetheine in the LasI pocket, and stabilizing polar contact between the ligand’s central amide group and Arg30 sidechain is expected to guide the long acyl ligand through the V-shaped cleft and deep into the elongated site. On the other hand, the significance of ligand-wise interactions with hydrophobic V-cleft residues, Phe27 and Trp33, was also rationalized since both residues are significantly important for recognizing the LasI substrate (S-adenosyl-L-methionine) and bringing it to the proper orientation for catalysis [65]. Further stabilization of the SIT and OMR docking poses was mediated through additional polar and hydrophobic interactions with the pocket-lining residues. The free amine of SIT and the tetrahydropyrane ring of OMR predicted favorable hydrogen bonding with the mainchain and/or sidechain of the Arg104 residue. On the other hand, close-distance π–H hydrophobic interactions were depicted between the Trp69 NH-sidechain and the anchored triazole- and pyrazole-based rings of SIT and OMR, respectively. The above preferential docking orientations of both SIT and LIN reasonably translated into comparable high negative docking scores (S = −6.8131 and −6.4685 Kcal/mol, respectively).

It is worth mentioning that validating the above *P. aeruginosa* LasI-based in silico study was highly rational since the PDB-deposited protein target lacked its native co-crystallized ligand. In this regard, a validation approach was performed by investigating the molecular docking simulation of a reported control ligand (TZD-C8) and one of the modest investigated antipseudomonal gliptins, trelagliptin (TRG). Not surprisingly, the TRG exhibited a less negative docking score (S = −5.2137 Kcal/mol) when docked at the LasI’s binding site compared to the top three active investigated gliptins. Having a different structural topology of an arrow-like configuration rather than an extended form, this gliptin member was unable to achieve significant anchoring in the pocket’s elongated tunnel (Figure 7C). The TRG’s fluorinated aromatic scaffold settled at the V-shaped cleft in a similar orientation to those of SIT and OMR. However, the close substitution for the 3-amino piperidine ring on the central pyrimidine-2,4-dione core scaffold (ortho-position) as well as the rigid cyano functionality at its meta-position would impose steric hindrance when the ligand was docked inside the elongated tunnel pocket. Thus, TRG orientation was restricted to the V-shaped cleft while being just at the entrance of the elongated hydrophobic tunnel. It is worth noting that even with the sole accommodation of the V-shaped cleft, TRG can still hamper the entrance of LasI natural substrates, conferring potential interference with the target biological activity. The stability of the TRG binding mode was mediated via polar interaction with the Arg30 sidechain, the mainchains of both Arg104 and Phe105, and π–H hydrophobic contacts with Phe27 and Trp33 aromatic rings.

Finally, the result of the molecular docking simulation of the positive control reference ligand was consistent with reported findings [69] and had a comparable binding mode to those of the top active antipseudomonal gliptins. This ligand was reported by Lidor et al. as a synthetic inhibitor of the *P. aeruginosa* LasI-type quorum-sensing signaling synthase and showed potent inhibition of biofilm formation, altered quorum-sensing signal production, and hampered swarming motility [69]. In the present docking study, the polar thiazolidine-2,4-dione scaffold showed polar interactions with Arg30 and Ile107, underscoring the significance of such polar residues for stabilizing the target in complex with ligands with great structural diversity (Figure 7D). The importance of the above interactions was confirmed by Lidor et al. with site-directed mutagenesis of Arg30 as well as Ile170- [69]. The TZD-C8-mediated LasI inhibition activity was abolished following double site-directed mutagenesis, obtaining R30D and I107S. Anchoring of the TZD-C8 hydrophobic flexible tail (unsaturated) in the elongated tunnel site was predicted since successful maneuvers would be adopted to minimize any potential steric clashes with the pocket’s lining residues. The above-described docking findings validate the adopted molecular docking protocol and the obtained docking scores. It is worth mentioning that the docking score of the reported TZD-C8 (S = −5.5102 Kcal/mol) was inferior to the top docked investigated gliptins (S = −6.4685 up to −7.1987 Kcal/mol), which should confer superiority and higher antipseudomonal potentiality for the investigated antidiabetic agents. Such a poor docking score could be due to the simplicity of the acyl hydrophobic tail, which should result in lower hydrophobic interactions with the lining residues as compared to the aromatic terminal tails possessed by the investigated gliptins.

##### Docking Analysis at QscR-Type QS Protein Binding Site

We also explored the mechanism of antipseudomonal biofilm inhibition activity of the top active gliptins by conducting in silico prediction of the drugs’ affinity towards the QscR quorum-sensing protein, which is responsible for regulating the transcription of several bacterial virulence genes. The active antipseudomonal gliptins, SIT and OMR, exhibited favorable contacts with several residues of the target’s pocket. Validation of the docking simulation workflow was performed through a triple approach design. First, the co-crystallized C_12_-HSL was redocked at the QscR binding site using the same parameters of the gliptin investigated in the docking simulation, resulting in a very low root-mean-square deviation (RMSD) of 1.0850 Å (Appendix A). The low RMSD clearly validates the use of the docking procedure to study the interactions between the gliptins and the protein [70]. 

The deposited QscR protein is co-crystallized with an autoinducer with a nanomolar activation potency (EC50 = 5 nM). Thus, we intended to use a reported *P. aeruginosa* QscR antagonist or even a weaker agonist as a control reference ligand within our second docking approach to evaluate the antagonistic potentiality of our investigated gliptins. The reported synthetic phenyl HSL analog, namely, Q9, illustrated a weak *P. aeruginosa* QscR agonism profile with an EC50 value above 70 nM [67]. This ligand has the N-acyl homoserine lactone scaffold with its acyl chain incorporating a central phenyl ring core and an extended terminal saturated aliphatic chain (octanyloxy group) at the para position. Notably, Q9 inhibited 80% of reporter activation with IC50 values in the mid-nanomolar range (26 nM) [71]. This activity identifies Q9 as the most potent known QscR inhibitor [71].

Interestingly, we observed common conformation/orientation binding poses for the top active investigated hypoglycemic agents (Figure 8A). The fluorinated aromatic hydrophobic features of both SIT and OMR nestled at the large-sized hydrophobic subsite with limited steric clashes. Unlike the hydrophobic subsite pocket that binds the hydrophobic portion of the ligands, the more polar portion (fused ring) of the ligands showed some variability in their interactions at the small subpocket of the protein. This is quite different from the observed binding of SIT and OMR at LasI’s binding pocket, clearly due to the differences in the binding pocket topology of LasI and QscR. While LasI has a narrow twisted V-shaped binding cleft, QscR, on the other hand, has an elongated extended pocket topology that allows for the binding of polar fused rings without significant steric clashes with the pocket’s lining residues. Additionally, the calculated Richard’s solvent-accessible surface area-volume for the QscR binding pocket was significantly larger than that of the LasI binding site (579.64 Å2 −331.18 Å3 vs. 496.61 Å2 −254.06 Å3).

For the simulated TRG molecule, it anchored its terminal polar functionalities (3-amino-piperdine moiety) at the QscR small subpocket while directing their terminal aromatic scaffolds towards the larger hydrophobic subpocket. The depicted orientation/conformation of the investigated gliptins predicted great superimposition of the ligand’s ionizable heads with the lactone ring of the synthetic inhibitor (Q9) as well as QscR’s co-crystalline ligand, C_12_-HSL. Notably, their respective aromatic scaffolds were directed towards the larger hydrophobic subpocket. The docked ligands at QscR exhibited more extended conformations as compared to those within LasI’s V-shaped pocket. This was most obvious for SIT, predicting an almost linear conformation of its aliphatic spacer (3-amino-butanoyl linker) extending its terminal fluorinated aryl and heterocyclic moieties at both far ends of the QscR subpockets. This observation could be reasoned for the larger and more extended pocket size assigned for QscR, where residues at the large hydrophobic subpocket also impose less steric hindrance for anchoring the ligand’s terminal aromatic groups.

Significant polar contacts with the anionic Asp75 sidechain were observed for almost all docked gliptins (Figure 8B–D). Asp75-mediated polar binding interaction with the ligand’s NH or tautomeric/poor electronegative (δ+) nitrogen atoms appeared to be significant for anchoring the ligand at QscR’s small subpocket. In contrast, TRG lacked relevant interaction with the charged Asp75 owing to its unfavored orientation within the binding pocket. Having both the fluorinated phenyl and 3-amino piperidine groups at ortho positions to each other imposed steric hindrance with the protein residues, causing a significant ligand movement away from Asp75 (Figure 8E). Further stabilization of the docked adrenoreceptor hits was mediated through a wide range of polar residues, including: Ser38 and Tyr66 for SIT, and Ser38 and Tyr66 for TRG. More extensive polar interactions were correlated with the best docking scores for OMR (S = −7.4661 Kcal/mol).

Aside from the SIT/Asp75 hydrogen bond pair, both SIT and TRG depicted similar hydrogen bond patterns to those seen with Ser38 and Tyr66 residues. Nevertheless, the latter ligand, TRG, showed less optimal hydrogen bonding parameters (Appendix A). This observation in part explains the better docking score for SIT as compared to TRG (S = −7.0082 vs. −5.6625 Kcal/mol). Interestingly, the investigated gliptins illustrated wider polar networks with QscR’s lining residues in relation to those of LasI’s binding pocket, which should serve as a crucial driving force for anchoring small molecules within the QscR binding pocket. It is worth noting that Ser38 and/or its vicinal residues, Tyr58 and Tyr66, showed the most conserved hydrogen bond pair with the investigated gliptins. Ligand interactions with Ser38 have been reported to be vital in determining QscR’s signal specificity, where it showed the preferential binding of 3-O-HSL over unsubstituted native ligands [72]. Thus, the observed significant affinity of the investigated gliptins towards the QscR pocket is in part due to hydrogen bonding with Ser38.

The hits also showed relevant van der Waals hydrophobic interactions with several QscR non-polar residues, including Phe39, Ala41, Tyr52, Tyr58, Trp62, Ile77, Leu82, Phe101, Trp102, Pro117, Ile125, and/or Met127 amino acids. Extended π-mediated non-polar interactions were also observed for stabilizing the ligand/QscR complexes, particularly through π–π interaction with Phe54 as well as CH–π contacts with Phe39, Tyr52, Phe54, Tyr66, and/or Trp90 (Appendix A). It is worth noting that TRG exhibited the least non-polar contacts with the pocket residues, particularly those at the larger hydrophobic subpocket (Phe39, Ala41, Tyr52, Leu82, Pro117, and Ile125). The latter added to the differential docking scores between SIT and TRG, where both polar and hydrophobic interactions were considered crucial for small-molecule binding to the QscR active site.

Similar to polar interactions, ligand/QscR non-polar interactions were more abundant than those at LasI, explaining the preferentially higher overall docking scores for the ligand/QscR binding. Additionally, hydrophobic van der Waals bonding with Arg42 sidechain hydrocarbons was also observed for all QscR-docked gliptins, except for TRG. A comparable residue-wise binding profile was illustrated for the positive control, the potent QscR inhibitor Q9, where its amidic lactone scaffold mediated polar contacts with Ser38, Tyr58, and Asp75 sidechains. Additionally, significant hydrophobic contacts with Phe54, Arg42, Tyr52, Tyr58, Tyr66, Ile77, Leu82, Pro117, Ile125, and Met127 sidechains via its aromatic/aliphatic lipophilic tail conferred its high negative docking score (S = −8.1451 Kcal/mol). Validation of the Q9 binding mode was assured through a very low RMSD value (0.8706 Å) relative to the co-crystallized ligand. The overall computational findings point to the important role of the above-mentioned QscR amino acid for anchoring the investigated hits within the QscR binding site, which may be critical for their translation into potential competitive inhibitors towards *P. aeruginosa*.

##### Docking Analysis at LasR-Type Quorum-Sensing Protein Binding Site

For comprehensive exploration of the antipseudomonal biofilm inhibition activity, a molecular docking simulation study of the gliptins and another quorum-sensing protein, LasR, was also performed. LasR is known to exhibit promiscuity towards several noncognate HSL autoinducers produced by several bacterial species [68]. This promiscuous behavior of LasR has been suggested to be related to the ability of *P. aeruginosa* to occupy niches of other HSL-producing bacterial species, giving them an advantage over such competitors [73]. We expected that the top active gliptins would competitively occupy the LasR binding site and prevent *P. aeruginosa* from surviving in environments with competing species [74]. The deposited LasR protein co-crystallized with the cognate autoinducer, 3-oxo-C_10_-HSL (EC50 = 15 nM), a potent LasR low-nanomolar-range agonist, the thing that provoked us to adopt the reported LasR antagonist for validating the docking/simulation study. The phenyl derivative HSL analog Q9, a potent QscR antagonist, was also reported with 10-fold higher antagonist potency than any LasR inhibitors [71]. Additionally, Q9 does not exhibit the atypical partial agonist characteristics of other LasR inhibitors at high concentrations [71]. The docking study with the known co-crystallized ligand resulted in a very low RMSD of 1.0576 Å, ensuring the biological significance of the subsequent docked gliptin poses (Appendix A).

The docked gliptins in the LasR active pocket showed comparable ligand orientations. The ligands’ polar heads perfectly anchored in the small subpocket of LasR with their nitrogen atoms superimposed closely with the amide group of the reference control ligand, Q9 (Figure 9A). The ligands’ terminal aromatic (phenolic or heterocyclic) scaffolds also fit in the large hydrophobic pocket of LasR with a similar orientation to Q9′s non-polar terminal acyl functionality. Although these ligands exhibited comparable orientations in the LasR binding pocket, their binding conformations differed when docked to QscR. Generally, gliptins at QscR adopted elongated, almost straight conformations with their terminal scaffolds extended at both far ends of the target pocket. However, the same ligands adopted a common crescent-like conformation for their anchored terminal aromatic scaffolds, orienting toward the opposing face of the ligand-binding pocket. This conformation could explain the comparably lower LasR binding site volume as compared to that of QscR (258.43 Å2 −120.24 Å3 vs. 579.64 Å2 −331.18 Å3). Despite such tight space in the LasR binding site, ligands managed to be accommodated in the pocket, furnishing moderate-to-high negative docking scores (S = −4.5989 to −7.1122 Kcal/mol) for the gliptins and even a far superior docking score for Q9 (S = −8.3188 Kcal/mol). We propose that the promiscuity of the LasR pocket could be the reason for such differential binding modes, where non-polar lining residues are of smaller sizes corresponding to those of the same sequence number in QscR. For instance, Ala127 and Val132 within the LasR hydrophobic subpocket correspond to QscR Met127 and Arg132, respectively. Those earlier small-sized residues would impose less sterically hindered paths against the anchoring of the gliptins’ terminal aromatic scaffolds within the LasR hydrophobic subpocket.

Polar interaction with the anionic charged residue, Asp73, was conserved across almost all docked gliptins, with hydrogen bond interactions formed with the Asp73 carboxyl oxygen atom(s). Additionally, an almost conserved strong hydrogen bond interaction with the Ser129 sidechain was observed (Appendix A). Extra hydrogen bond interactions were also observed for the Trp60 sidechain εN-atom with SIT, while OMR and TRG ligands showed hydrogen bond interaction with the Tyr56 sidechain (Figure 9B–D). All docked gliptins showed favorable hydrogen bond interactions with the sidechain of Arg61. A comparable polar interaction pattern was observed for the reference inhibitor, Q9, with its lactone head and amidic group forming favorable polar interactions with Tyr56, Asp73, Tyr93, and Ser129 (Figure 9E).

Accommodation of the ligands’ non-polar scaffolds at the large hydrophobic site of LasI is expected to play an important role in stabilizing the gliptins at the LasI target pocket. Comparable hydrophobic contacts with several residues, including Ile52, Trp60, Tyr93, Phe101, Ala105, Leu110, and Ala127, were observed for all investigated gliptins as well as the positive control ligand, Q9. In fact, the tightness of LasI’s pocket enabled close contacts between the gliptins and hydrophobic amino acids. Further stability of the ligand–LasI complexes occurred through other hydrophobic and π-driven interactions with the non-polar residues of Leu40, Tyr47, Ile52, Phe101, Leu125, and Ala127 at the terminal end of LasI’s hydrophobic pocket. Nevertheless, such extra stability was only permissible for ligands exhibiting an extended conformation, allowing them to extend to the end of LasI’s subpocket. Only SIT, OMR, and Q9 managed to exhibit close-range hydrophobic contacts with Leu40 and Leu125 as well as π–π and/or π–CH non-polar interactions via their terminal hydrophobic groups (Appendix A). These terminal non-polar interactions at the target’s hydrophobic subpocket are expected to satisfy the hydrophobic potentiality of the lining residues. These observations could explain the high negative scores of these compounds (S = −7.1122, −6.4583, and −8.3188 Kcal/mol for SIT, OMR, and Q9, respectively).

In contrast, the non-extended arrowhead conformation of TRG imposes steric hindrance for the ligand binding at LasI’s hydrophobic pocket (Figure 9E), similar to the TRG docking pose with QscR. This could explain the modest docking binding score (S = −4.5989 Kcal/mol). It is clear that SIT, OMR, and Q9 share a similar hydrophobic binding pattern with LasI’s non-polar residues. We propose that differential docking scores among these top ligands could be due to differences in the extent and magnitude of polar (hydrogen bond) interactions rather than through hydrophobic interactions. On the other hand, the differential docking scores for the top active gliptins in relation to the modestly active one, TRG, are likely due to the ligands’ terminal aryl groups rather than their polar scaffolds at LasI’s small subpocket. Finally, the in silico findings suggest that the above-mentioned pocket residues play an important role in stabilizing the gliptin within the target binding site and provide insight into their competitive inhibition activity in relation to Q9′s binding mode.

##### Docking Simulation on *S. aureus* Virulence-Regulating Biotargets

We also conducted a validated molecular modeling simulation study of our top antivirulence gliptins (SIT and TRG) against two targets of the *S. aureus* prototypic two-component signaling (TCS) system: the sensor histidine kinase cognate protein (AgrC) and response regulator (AgrA), which regulate the virulence factors, biofilm formation, antibiotic resistance, and the expression of toxin genes within myriad pathogenic bacterial species [75,76]. The *S. aureus* survival-driven system comprises the membrane-bound AgrC, which, upon its activation via extracellular signals, undergoes autophosphorylation on a conserved histidine amino acid. Phosphorylation further elicits intracellular signaling by activating the second cognate protein, a cytoplasmic AgrA, which is a transcription factor driving the accessory gene regulator (agr) of QS gene transcription [77]. The transferred phosphate group from AgrC enhances the AgrA–DNA binding through dimerization for altered gene expression towards bacterial survival responses against changing environments [78]. Targeting TCS has been considered promising for identifying new safe antibacterial compounds since this system is ubiquitous in bacterial species but absent in mammals [79,80]. TCS is also implicated in several types of bacterial-related antibiotic resistance, including vancomycin, carbapenem, and multi-drug resistance in *S. aureus*, *P. aeruginosa*, and Mycobacterium tuberculosis, respectively [81,82,83]. Despite TCS’s varied functions, these proteins share a catalytic core, and some proteins are even intimately coupled so that exploring multi-targeted therapeutics would deactivate multiple TCSs simultaneously, globally minimize TCS-driven signaling, and finally lead to greater bacterial damage [84,85,86].

##### Docking Analysis at ArgC ATP-Binding Domain

The 17.44 kDa ATP-binding domain of ArgC histidine kinase (PDB ID: 4BXI) at 2.20 Å resolution [78] was used for the molecular docking study. *S. aureus* AgrC is a 430-residue obligate dimer protein with a membrane-embedded sensor module and cytoplasmic histidine kinase module. The latter comprises a catalytic ATP-binding domain and dimerization/histidine-phosphotransfer domain harboring the phosphoacceptor His239 residue [87]. We focused our study on the ATP-binding domain (278–430 residues) because of its high conservation across variable TCSs, while this structure architecture is absent from the human-abundant tyrosine and serine/threonine kinases and mammalian histidine kinases [88,89]. Thus, targeting the AgrC ATP-binding domain should confer selectivity and permit effective and safe bacterial infection management within human hosts [84]. This strategy has been found to be useful in several reported studies in which designing small molecules for ATP-binding site competitive binding enabled multiple histidine kinases’ inhibition within different bacterial species [84,90,91,92,93,94,95,96,97,98].

Structurally related to the GHL/ATPase family, the ArgC ATP-binding domain adopts a distinct Bergerat fold in the form of sandwiched α-helices (α1-4) and mixed β-strands (β1-6) for the two layers, along with a flexible/discrete ATP-lid segment covering the nucleotide-binding pocket (Figure 10A) [78]. The Bergerat fold is composed of five homology boxes (G1-3, F, and N boxes) with residues that are pivotal for ATP recognition and binding. The conserved G1-box aspartate residues (Asp373 and Asp374) mediate polar interaction with ADP’s endo/exocyclic N-atom, while the N-box asparagine residue (Asn339) allows the stability of ADP through Mg2+ metal-bridge polar interactions with the phosphate group [88]. Positioning the adenine ring within the ATP-binding site is further achieved through hydrophobic interactions as well as π-mediated ring stacking with residues of the F, G1, and/or G3 boxes, including: Met368, Ile375, Leu381, Phe382, Thr414, Ile415, Ile416, Phe420, and Phe421 (Figure 10B). Connecting the F and G2 boxes, the ATP lid is generally tethered between two lipophilic patch motifs and possesses a conserved arginine residue (Arg393) that interacts with nucleotide β-phosphates, suggesting its catalytic role [87]. The above pocket residues in AgrC-corresponding histidine kinases (*S. epidermidis* YycG, *S. pneumoniae* VicK–HK853, *Streptococcus mutants* VicK–CiaH–LiaS, and Shigella flexneri PhoQ histidine kinases) were depicted as significant for anchoring variable small ligands for competitive binding at the ATP-binding site [91,92,93,94,95,96,97,98]. Several of these studies correlated the ligand–target affinity to significantly low micromolar in vitro inhibition [92,96,97,98] and antibacterial activities [91,92,97,98], and/or they were found to be promising within animal models [93,94,95].

Despite the fact that AgrC can be crystallized in its holo state (i.e., bound with ATP), none of the crystallized AgrC PDB files deposited within the protein data bank revealed ATP electron density [78,87]. The co-crystal structure of ADP-β-N and similar ATPase proteins (PDB ID: 1I59 and 2C2A) are known. In addition, OMR, which is a modest anti-staphylococcal gliptin complex structure with ATPase, is known. We therefore used these ligands as controls to validate modeling studies. Interestingly, the molecular docking simulation of the investigated gliptins showed a common orientation of these docked compounds within the AgrC ATP-binding site (Figure 10C). The investigated ligands showed deep anchoring via their aromatic heterocyclic rings through the space between α2 and α3 while reaching down towards the β4-6-sheet sandwich interface. The ligands’ heads were docked in near superimposition with the adenine ring of the AgrC ADP-β-N substrate in proximity to three homology boxes (G1, G3, and N) of the Bergerat pocket fold. On the other hand, the tails of the docked gliptins, comprising their central aliphatic linker/ring spacer as well as terminal decorated rings, were directed towards the solvent side and adjacent to the F homology box at the α3 helix. Unlike the obtained gliptins’ binding modes, the docked ADP-β-N showed more pocket confinement within the ATP-binding site, where it stretched across whole homology boxes and reached down to the far G2 via its terminal phosphate groups. This depicted differential binding mode was mostly related to the smaller size of ADP-β-N as compared to the investigated gliptins.

The latter compounds can generally adopt more extended conformations, which would cause significant steric clashes with the pocket G2 box/α4 helix residues. Conformation analysis of the crystallized AgrC protein illustrated that both α3 and α4 positions are more likely to hinder large substrate binding due to their relative conformations and their presence at the active site entrance [78]. In these regards, large-sized gliptins, such as SIT and OMR, showed minimal and nearly no significant proximity to the G2 homology box. It is worth mentioning that TRG is smaller compared to other investigated gliptins; however, this ligand also lacked significant contacts with the far G2 box. This can be explained by the fact that TRG’s chemical structure comprises a central pyrimidin-2,6-dione ring harboring two bulky substitutions at positions 3 and 5, which cause this small ligand to adopt an arrowhead/V-shaped conformation within the AgrC active site. However, due to the tightness of the pocket imposed by the gating α3/4 helices, the TRG arrowhead conformation was directed so that only one of its side arrow vectors (cyano-fluorinated benzene) was anchored at the adenine-binding site, with the more polar ring (3-amino piperidine) directed towards the bulk solvent.

Investigating the differential gliptin binding modes showed that the top docked ligands with almost comparable high docking scores (S = −5.2235 and −4.9504 kCal/mol for SIT and TRG, respectively) formed extended polar contacts with key pocket residues (Appendix A). These ligands suggested significant polar contacts with the conserved G1 Asp374 residues. The central free amine linker of SIT and poor electronegative N atoms at TRG’s pyrimidine-2,6-dione ring suggested double polar contacts or even a salt bridge with the Asp374 sidechain (Figure 10D,E). Additionally, both the SIT and TRG ligands illustrated significant hydrogen bond pairing with the Asp374 mainchain NH peptide. Deep anchoring and better orientation of the SIT aromatic head should result in favorable ligand stability near the N box through polar interaction with the sidechain of the vicinal residue, Gln423. On the other hand, SIT and TRG were the only top docked gliptins that managed to exhibit significant polar contacts with any of the ATP-lid residues. The favored orientation of the ligands’ terminal rings permitted relevant polar contact with the Ser387 of this flexible polypeptide chain covering the nucleotide-binding site. Further ligand–pocket stability was depicted regarding the compounds’ terminal polar rings, where the 3-amino-substituted piperidine ring in TRG managed to form significant hydrogen bonding with Arg377 at the pocket entrance towards the solvent side. Besides the polar interactions, the top docked ligands exhibited hydrophobic contacts with key pocket residues, including Cys371, Ile375, Leu381, Phe382, Leu397, Ile415, and/or Phe421. Nevertheless, only SIT, which exhibited deeper pocket anchoring, managed to achieve proximity to G3-box Phe421 as well as van der Waal interactions with the sidechain carbons (Cβ) of the N-box Asn339 residue.

OMR showed a very low docking score (S = −3.7695 kCal/mol), which is likely due to significantly fewer polar contacts with the ATP-binding site. Furnishing hydrogen bonds with conserved G1-box Asp373 and Asp374 mainchain NH groups and a single polar contact with the ATP-lid Glu391 residue permitted relative ligand stability inside the active site (Figure 10F). Despite the proximity of OMR’s terminal fluorinated ring to the Arg377 residue, no relevant polar interaction was observed, which may explain the coplanar/parallel orientation of the ligand’s ring in relation to the Asrg377 arginine sidechain. It is worth mentioning that the ligand’s sulfone group permitted anchoring at the G1 box; however, such a polar moiety would impose a great electrostatic penalty being so close to the box Phe421 and Ile415. That orientation would compromise the anchoring of the ligand in the binding pocket, explaining the low docking score.

On the contrary, ADP-β-N showed highly extended polar networking with the pocket residues, translating into a significantly high docking score (S = −6.0269 kCal/mol). Polar contacts with the Asp373 mainchain and Gln423 sidechain are important in anchoring ADP-β-N at the nucleotide-binding site to involve several polar contacts between the ligand’s phosphate groups and mainchains of Ser387, Gly392, Arg393, and Leu397 residues (Figure 10G). Besides the polar interactions, the adenine ring is stabilized by sandwiching between Phe382 and Leu397 sidechains, as well as π–H hydrophobic contact with Phe421 and van der Waal interactions with Ile341 and Leu381. Notably, the docked ADP-β-N was capable of reproducing the reported key substrate–histidine kinase interactions, validating the adopted molecular docking protocol for predicting reliable ligand–target poses with successful translation into biological significance.

##### Docking Analysis at ArgA Cytoplasmic Response Regulator

The 22.33 kDa uncharacterized DNA-binding domain (LytTR) of *S. aureus* cytoplasmic response regulator AgrA (PDB ID: 3BS1, at 1.60 Å resolution) [90] was used to evaluate the gliptins’ potential to slow the *S. aureus* virulence transcription machinery. The AgrA-LytTR domain crystalized as an elongated β/β/β anti-parallel sandwich of two-fold symmetry across each of the five successive β-sheets (β1-5 and β6-10), with a two-turn short α-helix being uninvolved in DNA binding. The AgrC-LytTR domain recognizes and binds to a consensus nine-DNA-base-pair sequence (5′-ACAGTTAAG-3′) through base-specific interactions driven by three β-loops (L1-3) protruding from the β-sheet edges (Figure 11A). The three loops bind to the DNA at its two successive major grooves (G1,3) with an intervening minor one (G2) [90]. Alanine mutagenesis studies revealed that three amino acids (His169, Asn201, and Arg233) are highly responsible for the specific AgrA-LytTR/DNA binding, each residue for one respective DNA groove. The three amino acids exhibit direct and/or water-mediated interactions with the DNA bases and/or ribose sugars (Figure 11B) [90,99].

Alanine-mutated AgrA showed lower DNA binding affinities with increased Kd values of almost 40-, 10-, and 90-fold for His169, Asn201, and Arg233, respectively, as compared to the wild-type protein [90]. A systematic alanine-based mutational analysis also revealed that four additional DNA-proximal residues (Leu171, Glu181, His200, and Tyr229) are pivotal for AgrA transcription activation while influencing the overall stability of AgrA ternary structure for DNA binding/distortion [100]. Nevertheless, this study showed the highly conserved amino acid, Tyr229, to have no negative impact on DNA binding following mutation. Based on the above evidence, the ability of small ligands to interact with these key AgrA-LytTR/DNA binding residues will be considered promising for hampering protein–DNA binding or additional transcriptional activation.

For a validated molecular docking study, a reference control inhibitor was adopted since none of the PDB-deposited AgrA high atomic resolution structures show bound ligands. Savarin (SAV) is one of the most cited *S. aureus* AgrA inhibitors for disrupting agr-driven quorum sensing [101,102,103,104,105]. This sulfone-substituted triazoloquinazoline compound enhances the host defense without an observed resistance/tolerance either in vitro or in vivo as compared to conventional antibiotics [101]. Additional docking of OMR, as one of the modest investigated anti-staphylococcal gliptins, would further validate the presented docking protocol. The in silico docking of SAV revealed the ligand anchoring at the space (Site I) across the L1 and L2 interfaces facing the DNA binding grooves and close to the protein’s carboxy terminus. As the target protein exhibits two-fold symmetrical architecture, another possible binding site (Site II) was also recognized during our initial pocket site analysis, close to the protein’s N-terminus across the L3 and L* loops and unknot involved in DNA binding. We decided to explore ligand docking at both sites to gain better insight regarding the ligand–AgrA binding interactions.

The docking results showed comparable orientations as well as conformations for the docked gliptins and the reference control at both sites (Figure 11C). The topology of both sites allowed the investigated ligand to adopt crescent-like conformations, although ligands at Site II showed more relaxed and extended conformations due to the larger cavity site size. This was confirmed through the MOE2019.01 geometry-based Site-Finder module, where 25 and 49 alpha spheres were assigned for Sites I and II, respectively. Alpha spheres represent geometrical features that can be used to map out concave interactions at protein pocket surfaces [106,107]. The pocket site differences could explain the differential ligand anchoring at both sites. However, preferable ligand binding at these two binding pockets could be revealed through comprehensive ligand–target interaction analysis.

In our docking study, the control reference SAV showed similar reported Site I–specific binding patterns, which have been depicted in several reported studies [101,102,105]. Polar interaction with Arg218, as well as a π-mediated hydrophobic contact with Tyr229, was illustrated in the presented docking study and in the current literature (Figure 11D) [101,102,105]. This could further validate our adopted docking protocol. The ligand was further stabilized through double hydrogen bonding with the key polar Arg233 residue, in addition to double π–H interactions between His200 and the triazole/pyrimidine rings, as well as Asn201 and the terminal phenyl ring (Appendix A). At Site II, SAV showed π–π contact with His169 and a single hydrogen bond with Lys187 of the two-turn α-helix. The latter binding profiles were translated into higher SAV docking scores at Site I as compared to Site II (S = −6.3532 vs. −4.8253 kCal/mol, respectively).

Similar Site I/Site II docking preferentiality was also illustrated for both the top active investigated gliptins (SIT and TRG) and the modest control, OMR. Higher and more extended polar interactions were observed for SIT at Site I, where the ligand’s triazolopiperazine ring managed to form contacts with the sidechains of Arg218 and Arg233, as well as Asn201 and Val232 mainchains (Figure 11E). Additional non-classical hydrogen bonding (π–H) was also observed with the His200 sidechain at Site I. On the other side, polar contacts with His169, Leu186, and Arg198, besides a π–H interaction with Ser164, were translated into a lower docking score at Site II (S = −5.1026 kCal/mol) compared to that at Site I (S = −6.6659 kCal/mol).

A comparison of the highest and lowest docked gliptins, TRG and OMR, showed that the docking scores appear to correlate with the extent of polar interaction networking as compared to hydrophobic contacts. Due to TRG’s non-linear topology, the ligand adopted radial proximity to the interface between Site I and Site II (Figure 11F). The TRG-Site I profile predicted double hydrogen bond pairing with the Arg233 side/mainchain, halogen non-classical bonding with His200, polar contact with the Asn201 sidechain, and π–π interaction between its pyrimidinedione ring and His200 sidechain (S = −6.9644 kCal/mol). A marginally lower docking score was obtained for TRG at Site II (S = −6.2761 kCal/mol), where polar contacts with His169, Gly184, and Leu186, as well as His169-driven π–π interaction, were predicted. Finally, the OMR showed polar contacts with Arg218 and Arg233 without relevant hydrophobic interactions with the surrounding Site I residues, translating into a modest docking score (S = −5.5185 kCal/mol) (Figure 11G). The latter binding mode could be correlated to the polar methyl sulfone functionality being oriented towards Tyr229, imposing repulsive forces that would orient the pyrazole ring far from the His200 interface. Despite the OMR π–π interaction with His169 at Site II, the ligand only exhibited a single polar contact with the same residue mainchain, furnishing much lower docking energy (S = −4.4773 kCal/mol).

Based on the modeling data, the top active gliptins are capable of anchoring at the AgrA-LytTR DNA-binding site within SAV, suggestive of a promising biological profile. Moreover, Site I- over Site II-docking preferentiality holds the advantage of interacting with a wider range of residues that are reported to be responsible for AgrA transcriptional and DNA-binding machinery. These include Arg233 and Asn201, which mediate specific direct DNA-binding contacts, as well as others, such as His200 and Tyr229, which are reported as being important for protein transcriptional activation and DNA affinity.

## 4. Conclusions

Counteracting bacterial resistance requires developing new innovative approaches, and targeting bacterial virulence is one of them. Mitigating bacterial resistance confers several merits, as it does not influence the bacterial growth or induce the bacteria to develop resistance and, at the same time, facilitates their eradication by immune cells. Gliptins are dipeptidyl peptidase-4 inhibitors (DPI-4) that are used to control blood glucose levels in diabetes mellitus. In the current work, the antivirulence activities of gliptins were evaluated against two bacterial models, *P. aeruginosa* and *S. aureus*. However, all gliptins showed significant antibiofilm activities; sitagliptin and omarigliptin showed the most significant biofilm inhibition. Sitagliptin was chosen for further antivirulence evaluation. Sitagliptin at sub-MIC significantly protected mice against *P. aeruginosa* and *S. aureus*. The anti-QS activity of sitagliptin was assessed, and it significantly downregulated *P. aeruginosa* QS-encoding genes and *S. aureus* QS- and biofilm-formation-encoding genes. A detailed in silico monocular docking study was conducted to test the ability of different gliptins to hinder the QS receptors in *P. aeruginosa* and *S. aureus*. The molecular docking demonstrated the ability of sitagliptin and omarigliptin to bind to different QS receptors. The current study evaluated the preliminary possibility of gliptins and other related chemical moieties to serve as antivirulence and anti-QS agents; however, further pharmacological and toxicological studies are necessary prior to the use of gliptins as adjuvants to traditional antimicrobial therapy.

## Figures and Tables

**Figure 1 biomedicines-10-01169-f001:**
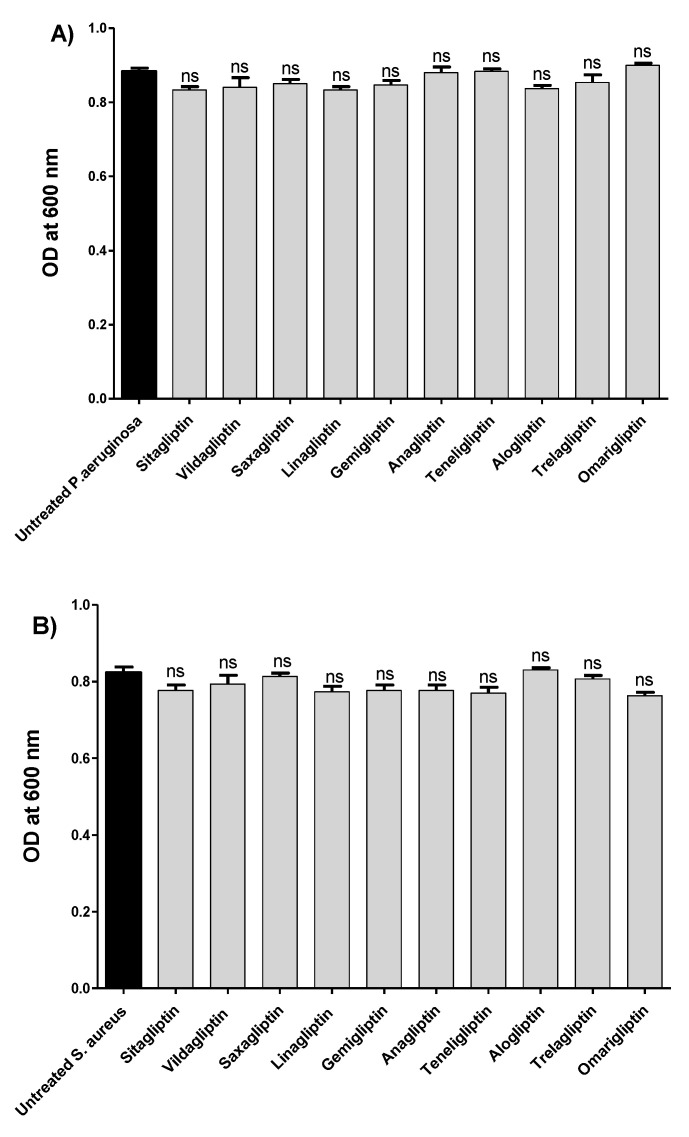
Effect of gliptins on bacterial growth. The turbidities of bacterial growth were measured at OD_600_ nm in the presence or absence of gliptins at sub-MICs (1/5 MIC). The experiment was repeated in triplicate, data are presented as mean ± SD, and two-way ANOVA test followed by Bonferroni post-test was employed to test significance. There was no significant effect of tested gliptins at their sub-MICs on the growth of (**A**) *P. aeruginosa* or (**B**) *S. aureus*. ns: non-significant.

**Figure 2 biomedicines-10-01169-f002:**
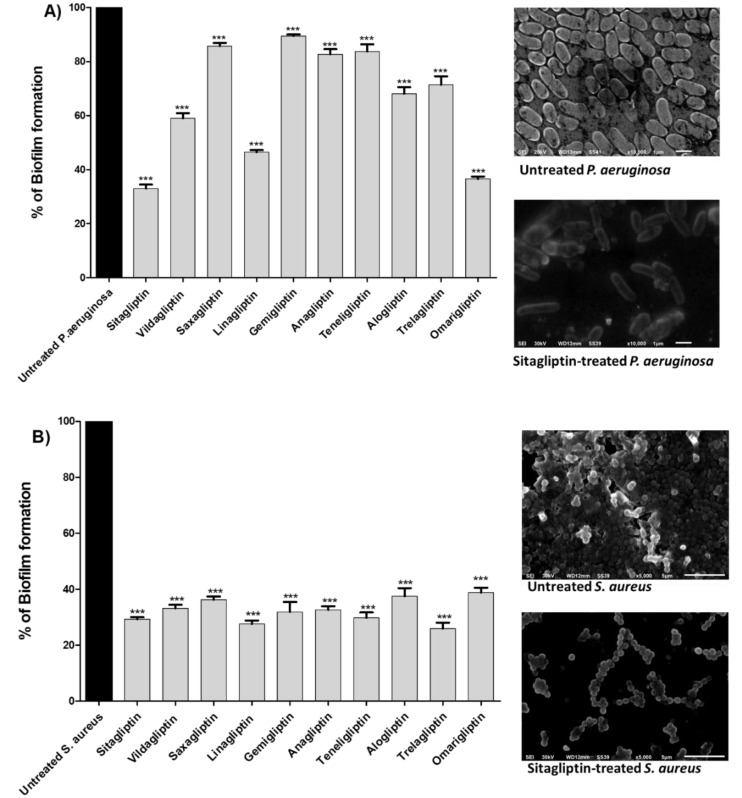
Antibiofilm activities of the tested gliptins. The crystal violet method was employed to stain the biofilm-forming cells in the absence or presence of gliptins at sub-MICs. The absorbances of the crystal-violet-stained biofilm-forming cells were measured at 590 nm in the absence or presence of gliptins at sub-MICs. The experiment was performed in triplicate, and the results are expressed as the percentage change from the untreated bacterial control. Data are presented as mean ± SD, and two-way ANOVA test followed by Bonferroni post-test was employed to test significance. Gliptins significantly diminished (**A**) *P. aeruginosa* or (**B**) *S. aureus* biofilm formation. Furthermore, representative electron microscope images were captured to show the effect of sitagliptin at sub-MIC on the formation of biofilm. Sitagliptin obviously inhibited both *P. aeruginosa* and *S. aureus* biofilms. ***: *p* ≤ 0.001.

**Figure 3 biomedicines-10-01169-f003:**
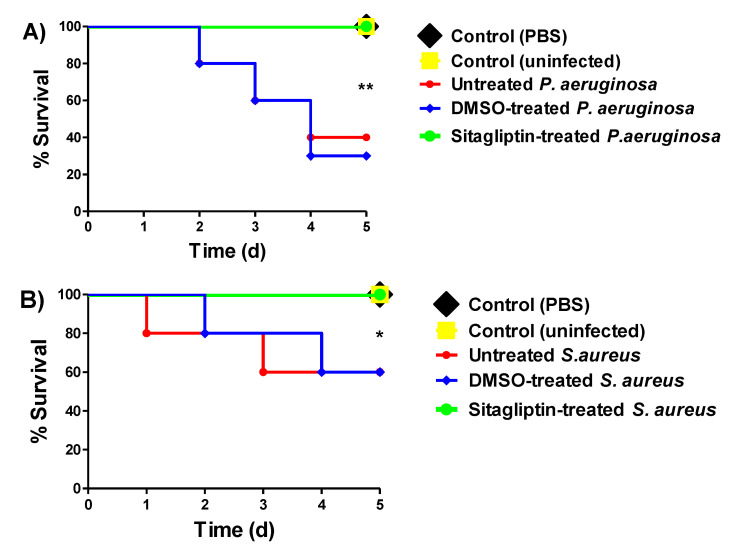
In vivo protection against *P. aeruginosa* and *S. aureus*. Ten mice groups were established to evaluate the antivirulence activity of sitagliptin at sub-MIC against *P. aeruginosa* and *S. aureus*. Five mice groups were intraperitoneally injected with untreated *P. aeruginosa*, DMSO-treated *P. aeruginosa,* sitagliptin-treated *P. aeruginosa*, or sterile PBS, and the last group was uninjected. The same grouping was used for evaluation against *S. aureus*: five mice groups were injected with untreated *S. aureus*, DMSO-treated *S. aureus,* sitagliptin-treated *S. aureus*, or sterile PBS or remained uninjected. The deaths among mice groups were recorded and plotted by Kaplan–Meier method, and log-rank test was employed to test the significance. (**A**) Protection against *P. aeruginosa*: sitagliptin protected all mice, in comparison to 3 deaths in the positive control groups that were injected with untreated *P. aeruginosa*, conferring 60% protection. (**B**) Protection against *S. aureus*: sitagliptin protected all mice, while two deaths were recorded in positive control groups that were injected with untreated *S. aureus*, conferring 40% protection. Sitagliptin showed significant reduction in the capacity of *P. aeruginosa* and *S. aureus* to kill mice; log-rank test *p* = 0.0028 and 0.0244, respectively. *: *p* ≤ 0.05, **: *p* ≤ 0.01.

**Figure 4 biomedicines-10-01169-f004:**
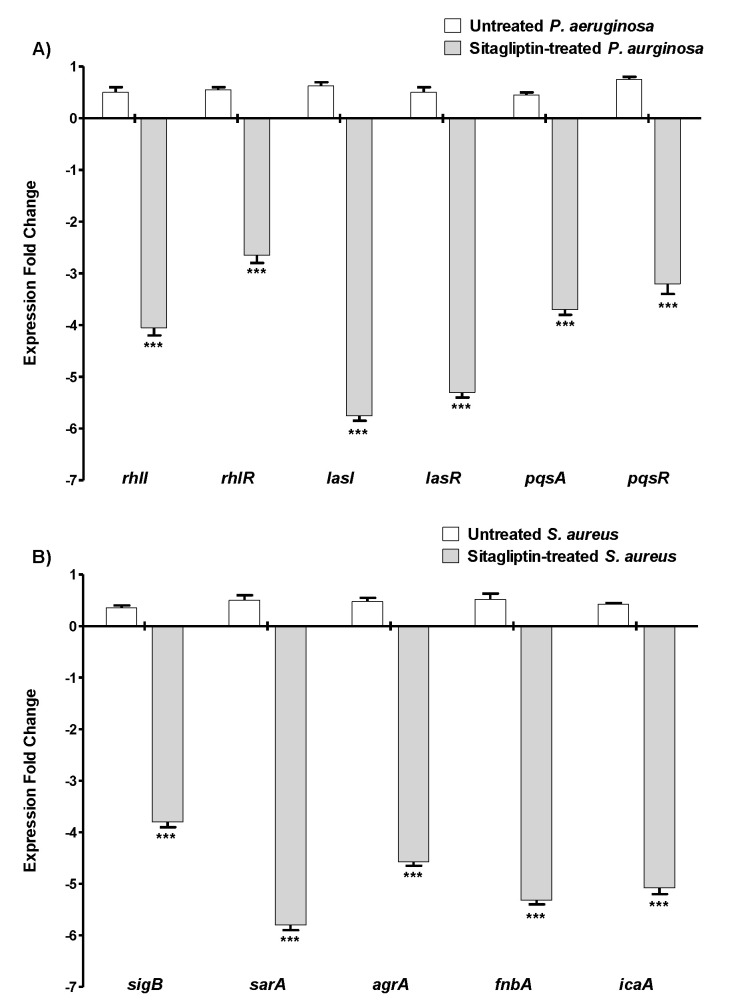
Downregulation of QS- and biofilm-formation-encoding genes in *P. aeruginosa* and *S. aureus*. RNA was collected from *P. aeruginosa* or *S. aureus* samples that were treated or not with sitagliptin at sub-MIC, and the expression levels of genes were normalized to housekeeping genes *rpoD* or *16s rRNA*, respectively. The experiment was repeated in triplicate, and the data are expressed as mean ± SD. One-way ANOVA test, followed by Dunnett’s multiple comparison test, was employed to test significance. Sitagliptin significantly downregulated the expression of (**A**) QS-encoding genes in *P. aeruginosa* and (**B**) QS- and biofilm-formation-encoding genes in *S. aureus.* ***: *p* ≤ 0.001.

**Figure 5 biomedicines-10-01169-f005:**
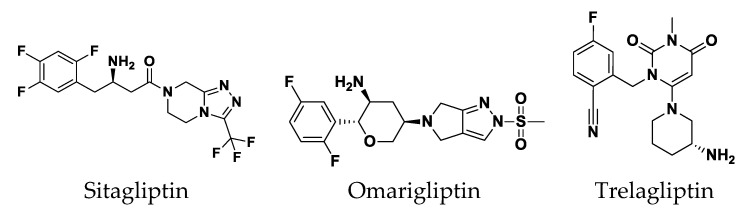
Two-dimensional structural representation of investigated gliptin drug members.

**Figure 6 biomedicines-10-01169-f006:**
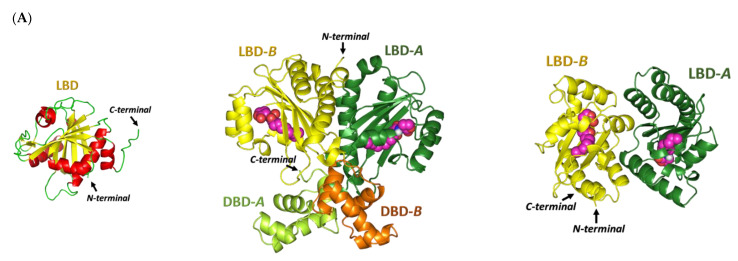
Architectures of three *P. aeruginosa* virulence-regulating biological targets. (**A**) Overall 3D cartoon representations of the *P. aeruginosa* LasI-type AHL synthase ligand-binding domain (PDB ID: 1R05; right panel), as well as QscR (PDB ID: 6CC0; middle panel) and LasR (PDB ID: 6MVN; right panel) quorum-sensing transcription proteins. Each quorum-sensing protein is colored differently in regard to its ligand-binding domain (LBD) and/or DNA-binding domain (DBD) as light/dark green and dark/light orange for protomers A and B, respectively. Ligands are represented as magenta spheres with the exception of LasI AHL synthase, where no ligand was co-crystallized with the protein. (**B**) Calculated putative pockets of the binding site topology at the three *P. aeruginosa* biological targets via the web-based Computed Atlas of Surface Topography of proteins (CASTp) server (http://sts.bioe.uic.edu/castp/index.html; accessed on 30 October 2021) using 1.4 Å radius probe, visualized as surface 3D representation, and colored in different colors (blue and red) for each target protomer. Estimated pocket area and volume were analytically calculated using Richard’s solvent-accessible surface model. (**C**) Surface representation of the binding site with an overlay of whichever co-crystallized ligand (magenta sticks) of its respective bacterial target is present: C_12_-HSL (6CC0) and 3-O-C_10_-HSL (6MVN). Hydrogen bonding is shown as dashed lines (red), while residues (green lines), only amino acids located within 5 Å radius distance from the bound ligand, are displayed and labeled in sequence numbers. Crystallized water molecule bridging the ligand/residue interactions is shown as red sphere.

**Figure 7 biomedicines-10-01169-f007:**
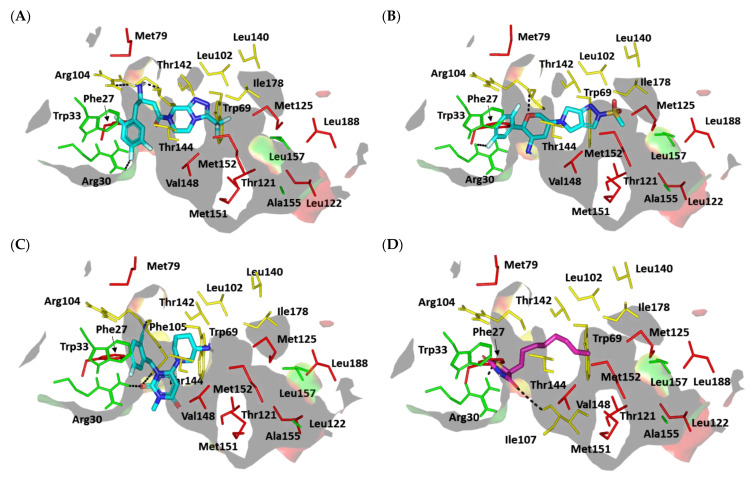
Ligand–protein binding interactions at *P. aeruginosa* LasI synthase binding site. Predicted binding modes of the docked ligands (cyan or magenta sticks): (**A**) SIT, (**B**) OMR, (**C**) TRG, and (**D**) TZD-C8. Only amino acids located within 5 Å radius distance from the bound ligand are displayed and labeled with sequence numbers. Non-polar hydrogen atoms are hidden for clarity. Hydrogen bonding is depicted as black dashed lines. (**E**) Overlay of investigated compounds (SIT, OMR, and TRG as cyan, green, and purple lines, respectively) and TZD-C8 (magenta sticks) binding to the protein’s canonical binding site comprising V-shaped cleft and elongated hydrophobic tunnel subpockets.

**Figure 8 biomedicines-10-01169-f008:**
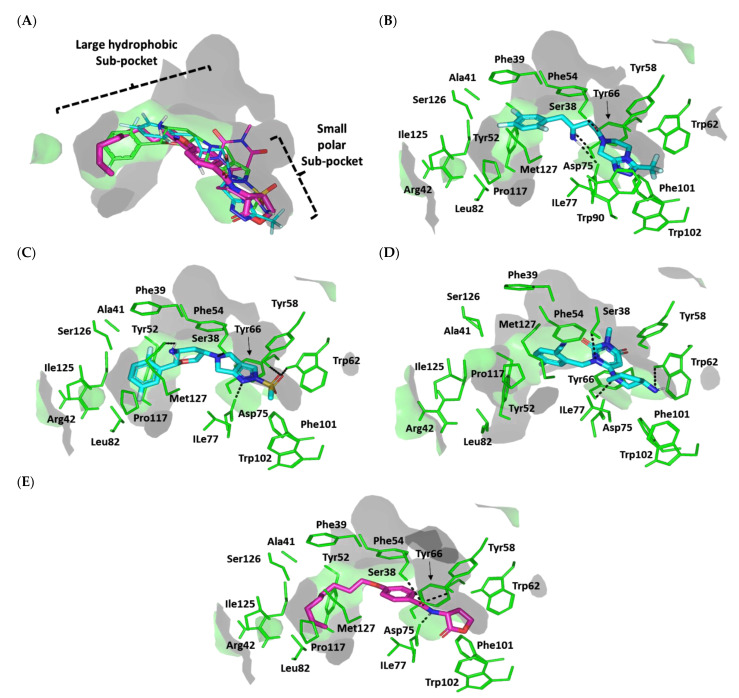
Ligand–protein binding interactions at *P. aeruginosa* QscR quorum-sensing transcription protein binding site. (**A**) Overlay of investigated compounds (SIT, OMR, and TRG as cyan, green, and purple lines, respectively) and Q9 (magenta sticks) binding to the protein’s canonical binding site comprising small polar head and larger hydrophobic subpockets. (**B**–**E)** Predicted binding modes of the docked ligands (cyan or magenta sticks): (**B**) SIT, (**C**) OMR, (**D**) TRG, and (**E**) Q9. Only amino acids located within 5 Å radius distance from the bound ligand are displayed and labeled with sequence numbers. Non-polar hydrogen atoms are hidden for clarity. Hydrogen bonding is depicted as black dashed lines.

**Figure 9 biomedicines-10-01169-f009:**
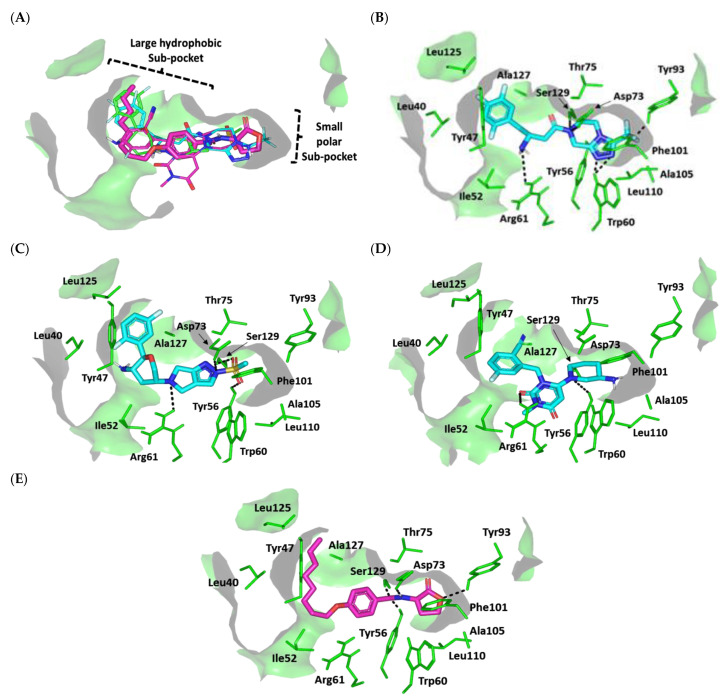
Ligand–protein binding interactions at *P. aeruginosa* LasR quorum-sensing transcription protein binding site. (**A**) Overlay of investigated compounds (SIT, OMR, and TRG as cyan, green, and purple lines, respectively) and Q9 (magenta sticks) binding to the protein’s canonical binding site comprising small polar head and larger hydrophobic subpockets. (**B**–**E**) Predicted binding modes of the docked ligands (cyan or magenta sticks): (**B**) SIT, (**C**) OMR; (**D**) TRG, and (**E**) Q9. Only amino acids located within 5 Å radius distance from the bound ligand are displayed and labeled with sequence numbers. Non-polar hydrogen atoms are hidden for clarity. Hydrogen bonding is depicted as black dashed lines.

**Figure 10 biomedicines-10-01169-f010:**
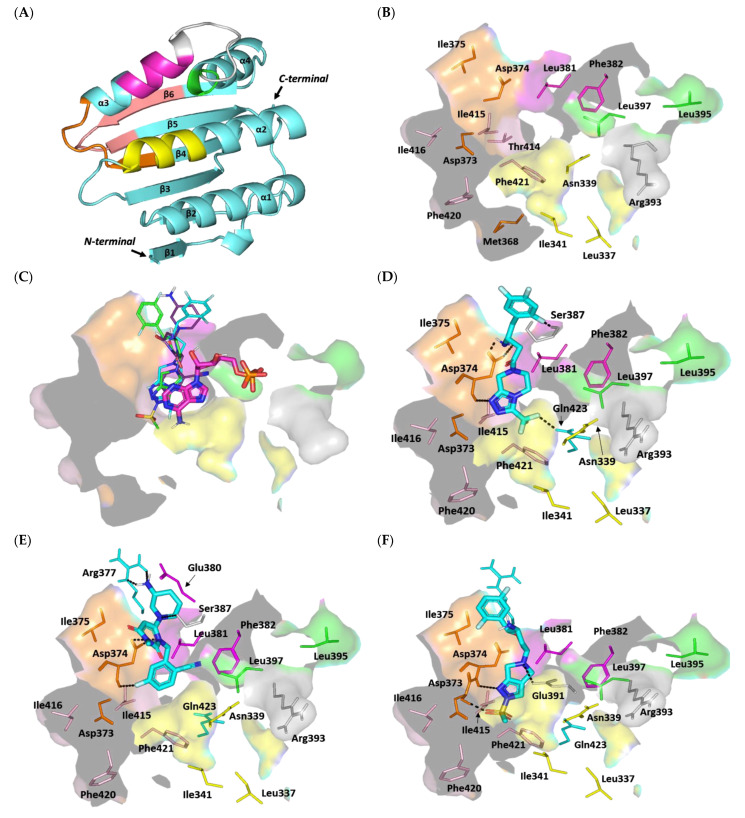
The architecture and predicted ligand–protein binding interactions at ATP-binding domain of *S. aureus* AgrC histidine kinase. (**A**) Overall 3D cartoon representations of the *S. aureus* AgrC ATP-binding domain (PDB ID: 4BXI) showing its distinct Bergerat folding in the form of sandwiched α-helices and mixed β-strands. The protein is colored differently in regard to its homology boxes and conservations: G1, G2, G3, F, and N boxes are illustrated in orange, green, pink, purple, and yellow, respectively. The ATP lid connecting G2/F boxes is represented in gray color. (**B**) Surface representation of the ATP-binding domain with key pocket residues (lines) involved in ADP binding. (**C**) Overlay of investigated compounds (SIT, OMR, and TRG as cyan, green, and purple lines, respectively) and ADP-β-N (magenta sticks) binding to the protein’s canonical binding site. (**D**–**G**) Predicted binding modes of the docked ligands (cyan or magenta sticks): (**D**) SIT, (**E**) TRG, (**F**) OMR, and (**G**) ADP-β-N. Residues are colored based on their respective homology boxes and labeled with sequence numbers. Non-polar hydrogens are removed for clarity. Hydrogen bonding is depicted as black dashed lines.

**Figure 11 biomedicines-10-01169-f011:**
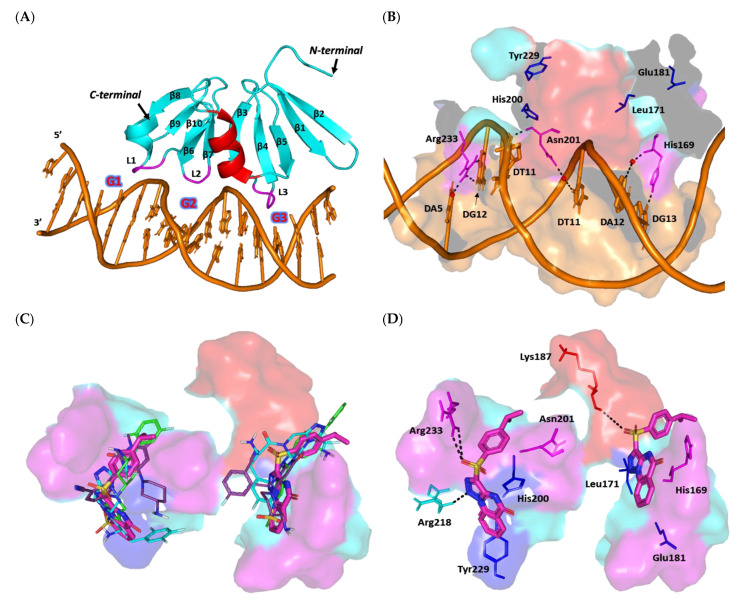
The architecture and predicted ligand–protein binding interactions at LytTR/DNA-binding domain of *S. aureus* AgrA cytoplasmic response regulator. (**A**) Overall 3D cartoon representations of *S. aureus* AgrA-LytTR/DNA binding sites (PDB ID: 3BS1) showing its single α-helix (red), two-fold symmetrical β-sheets (cyan), and three distinct loops (magenta) fitting in the major G1/minor G2/major G3 grooves of the co-crystallized DNA (orange). (**B**) Surface representation of the AgrA-LytTR/DNA-binding domain with key pocket residues (lines) involved within DNA binding/recognition (magenta) and/or transcriptional activation (blue). (**C**) Overlay of investigated compounds (SIT, OMR, and TRG as cyan, green, and purple lines, respectively) as well as SAV (magenta sticks) binding to the protein’s canonical binding site. (**D**–**G**) Predicted binding modes of the docked ligands (cyan or magenta sticks): (**D**) SAV, (**E**) SIT; (**F**) TRG, and (**G**) OMR. Residues are colored based on their respective sites and labeled with sequence numbers. Non-polar hydrogens are removed for clarity. Hydrogen bonding is depicted as black dashed lines.

**Table 1 biomedicines-10-01169-t001:** Sequences of the used primers for tested genes.

Target Gene	Sequence (5′–3′)	Reference
*lasI*	For: CTACAGCCTGCAGAACGACA Rev: ATCTGGGTCTTGGCATTGAG	[16,25,47]
*lasR*	For: ACGCTCAAGTGGAAAATTGG Rev: GTAGATGGACGGTTCCCAGA	[16,25,47]
*rhlI*	For: CTCTCTGAATCGCTGGAAGG Rev: GACGTCCTTGAGCAGGTAGG	[16,25,47]
*rhlR*	For: AGGAATGACGGAGGCTTTTT Rev: CCCGTAGTTCTGCATCTGGT	[16,25,47]
*pqsA*	For: TTCTGTTCCGCCTCGATTTC Rev: AGTCGTTCAACGCCAGCAC	[16,25,48]
*pqsR*	For: AACCTGGAAATCGACCTGTG Rev: TGAAATCGTCGAGCAGTACG	[16,25,47]
*rpoD*	For: GGGCGAAGAAGGAAATGGTC Rev: CAGGTGGCGTAGGTGGAGAAC	[48]
*SigB*	For: AAGTGATTCGTAAGGACGTCT Rev: TCGATAACTATAACCAAAGCC T	[49,50]
*SarA*	For: TCTTGTTAATGCACAACAACGTAA Rev: TGT TTG CTT CAG TGA TTC GTT T	[51]
*AgrA*	For: GGAGTGATTTCAATGGCACA Rev: ATCCAT TTTACTAAGTCACCGATT	[51]
*fnbA*	For: AACTGCACAACCAGCAAATG Rev: TTGAGGTTGTGTCGTTTCCTT	[51]
*icaA*	For: CAATACTATTTCGGGTGTCTTCACTCT Rev: CAAGAAACTGCAATATCTTCGGTAATCAT	[52]
16S rRNA	For: TGT CGT GAG ATG TTG GG Rev: CGA TTC CAG CTT CATGT	[50]

**Table 2 biomedicines-10-01169-t002:** MICs of gliptins for *P. aeruginosa* (ATCC 27853) and *S. aureus* (ATCC 6538).

Tested Gliptin	*P. aeruginosa*	*S. aureus*
Sitagliptin	10 mg/mL	5 mg/mL
Vildagliptin	10 mg/mL	5 mg/mL
Saxagliptin	2 mg/mL	1 mg/mL
Linagliptin	10 mg/mL	5 mg/mL
Gemigliptin	2 mg/mL	1 mg/mL
Anagliptin	2 mg/mL	1 mg/mL
Teneligliptin	2 mg/mL	1 mg/mL
Alogliptin	2 mg/mL	1 mg/mL
Trelagliptin	2 mg/mL	1 mg/mL
Omarigliptin	2 mg/mL	1 mg/mL

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
