# Peer review of "Anti-Quorum Sensing Activities of Gliptins against Pseudomonas aeruginosa and Staphylococcus aureus"

_biomedicines, 2022, doi:10.3390/biomedicines10051169_

Round 1

Reviewer 1 Report

Bacterial resistance to traditional antibiotics causes huge risks to public health. Anti-virulence agents are promising for these bacterial treatments. In this study, the authors used the approved drugs for diabetes treatment to treat pathogenic bacteria Pseudomonas aeruginosa and Staphylococcus aureus. Among them, sitagliptin protected mice from P. aeruginosa and S. aureus infections and down-regulated the QS-encoding genes in P. aeruginosa and S. aureus. In addition, a molecular docking study was conducted to evaluated the binding affinities of quorum sensing receptors. The study is significant. Some questions are below.

  1. The concentrations of the drugs used are high, even using 1/4 MIC. Therefore, the authors need to discuss the potential side effects and application prospect.
  2. Many figures from figs. 6-11 and words are used to evaluate the binding between the drugs and quorum sensing receptor. These results are predicted. Is there any experimental evidence to approve these results?
  3. Figure 1 showed only the growth at the last time point. It is more significant to show the growth curves at different time points.
  4. Figure 5 is not your results.
  5. The electron microscope images need more description in this manuscript.
  6. Why was not QscR tested in the RT-PCR experiments?
  7. Some words need italic. For example, line 170 in-vivo; line 379 P. aeruginosa,
  8. The genus names need For example, lines 131 and 132, Pseudomonas and Staphylococcus…
  9. 16s rRNA in table 2 does not need to be italic.
  10. There is updated Clinical Laboratory and Standards Institute Guidelines
  11. Some numbers need superscript or subscript. For example, line 166 10*8 CFU/mL, OD600, C12-HSL…
  12. The units need to be consistent. For example, mL and ml.

Author Response

Dear Reviewer,

We appreciate the reviewer for your valuable constructive comment, which greatly helped us improve the manuscript.

Please find the attached file of the response to all the raised points.

Wael

Reviewer 2 Report

In the paper entitled "Potential anti-virulence and anti-quorum sensing activities of gliptins against Pseudomonas aeruginosa and Staphylococcus aureus", the authors present in detail the anti-QS activities of antidiabetic gliptins, refering mostly to sitagliptin.

The research idea is very interesting and important for public health, considering the rapid development of bacterial resistance.

In addition, the paper is well structured and written. However, before acceptance in Biomedicines journal, it needs a minor to major revision:

  1. I would change the title to "Anti-quorum sensing activities of gliptins against Pseudomonas aeruginosa and Staphylococcus aureus" or to "Antivirulence activities of gliptins against Pseudomonas aeruginosa and Staphylococcus aureus through/by/using a and anti-quorum sensing mechanism", since an anti-QS is already an antivirulence one;
  2. replace anti-virulence with antivirulence;
  3. page 1 - line 35: replace "diminished the biofilms" with "diminished biofilm formation";
  4. page 2 - line 95: replace "compounds, sitagliptin" with "compouns. Sitagliptin";
  5. replace "in-silico" with "in silico";
  6. the concentrations used for the antivirulence activity investigation is MIC/5 (lines 126-129) or MIC/4 (line 241);
  7. replace "in-vivo"and "in-vitro with "in vivo" and "in vitro", respectively;
  8. line 164: replace "ability of tested gliptins" with "ability of the tested gliptins";
  9. define all abbreviations when first used within the text (eg. ip - line 172);
  10. "via" shoould be in Italic face;
  11. considering the MIC values determination, I believe it is advisable to have a reference drug (in this case, ceftazidime or imipenem maybe), in order to (more) correctly interpretate the antibacterial potential of the gliptins tested;
  12. line 265: replace "30% to 88" with either "30 to 88%" or "30% to 88%";
  13. line 268: "replace "A representative ....images" with "Representative ....images";
  14. line 282: replace "inhabited" with "inhibited";
  15. paragraph 3.3. - page 9: English language is poor: replace"was recorded over five days" with "was recorded for over five days" (line 305); replace "two mice...in the groups that was injected" with "two mice...in the groups that were injected" (line 306); lines 307: replace "While," with "Meanwhile,";
  16. decide if figure numbers mentioned within the text are Bold faced (Figure 2) or not;
  17. replace "Serratia" with "Serratia" or "Serratia ssp.";
  18. line 338: replace "synthesis authoinducers" with "synthesize autoinducers'
  19. lines 341-342: there are 2 or 3 QS systems?;
  20. line 379: P. aeruginosa should be Italic faced;
  21. in lines 380-381, the authors mention investigating and discussing the anti-pseudomonal activities of 3 gliptins (figure 5), while on line 432 they only mention 2 of them.

Author Response

(The authors gave the same response as above.)

Round 2

Reviewer 1 Report

Some minor revisions are needed.

Line 115, Pseudomonas is P.; Staphylococcus is S.

In table 1, 16s rRNA is 16S rRNA.

The table order is confused. Table 2 is first appeared. 

Author Response

Dear Reviewer, 

We highly appreciate the reviewer's interest in our manuscript and for valuable constructive comments, which greatly helped us improve the manuscript.

All the needed corrections are done (blue highlighted), please find attached the reply and revised manuscript.

Reviewer 2 Report

In this revised form of the manuscript now entitled "Anti-quorum sensing activities of gliptins against Pseudomonas aeruginosa and Staphylococcus aureus", the authors made all the changes advised/requested, improving thus the quality of their paper.

Author Response

Dear Reviewer, 

We highly appreciate the reviewer's interest in our manuscript and for valuable constructive comments, which greatly helped us improve the manuscript.

Thank you very much